# Intermittent Use of Portable NIV Increases Exercise Tolerance in COPD: A Randomised, Cross-Over Trial

**DOI:** 10.3390/jcm8010094

**Published:** 2019-01-15

**Authors:** Ioannis Vogiatzis, Nikolaos Chynkiamis, Matthew Armstrong, Nicholas D. Lane, Tom Hartley, William K. Gray, Stephen C. Bourke

**Affiliations:** 1Department of Sport, Exercise and Rehabilitation, Northumbria University, Newcastle Upon-Tyne NE1 8ST, UK; nikolaos.chynkiamis@northumbria.ac.uk (N.C.); matthew.armstrong@northumbria.ac.uk (M.A.); 2Northumbria Healthcare NHS Foundation Trust, Respiratory Medicine, North Tyneside & Northumberland NE29 8NH, UK; Nicholas.Lane@nhct.nhs.uk (N.D.L.); tom.hartley@nhct.nhs.uk (T.H.); william.gray@nhct.nhs.uk (W.K.G.); stephen.bourke@nhct.nhs.uk (S.C.B.); 3Institute of Cellular Medicine, Newcastle University, Newcastle Upon-Tyne NE1 7RU, UK

**Keywords:** non-invasive ventilation, COPD, exercise tolerance, pulmonary rehabilitation

## Abstract

During exercise, non-invasive ventilation (NIV) prolongs endurance in chronic obstructive pulmonary disease (COPD), but routine use is impractical. The VitaBreath device provides portable NIV (pNIV); however, it can only be used during recovery. We assessed the effect of pNIV compared to pursed lip breathing (PLB) on exercise tolerance. Twenty-four COPD patients were randomised to a high-intensity (HI: 2-min at 80% peak work rate (WRpeak) alternated with 2-min recovery; *n* = 13), or a moderate-intensity (MOD: 6-min at 60% WRpeak alternated with 2-min recovery; *n* = 11) protocol, and within these groups two tests were performed using pNIV and PLB during recovery in balanced order. Upon completion, patients were provided with pNIV; use over 12 weeks was assessed. Compared to PLB, pNIV increased exercise tolerance (HI: by 5.2 ± 6.0 min; MOD: by 5.8 ± 6.7 min) (*p* < 0.05). With pNIV, mean inspiratory capacity increased and breathlessness decreased by clinically meaningful margins during recovery compared to the end of exercise (HI: by 140 ± 110 mL and 1.2 ± 1.7; MOD: by 170 ± 80 mL and 1.0 ± 0.7). At 12 weeks, patients reported that pNIV reduced anxiety (median: 7.5/10 versus 4/10, *p* = 0.001) and recovery time from breathlessness (17/24 patients; *p* = 0.002); 23/24 used the device at least weekly. pNIV increased exercise tolerance by reducing dynamic hyperinflation and breathlessness in COPD patients.

## 1. Introduction

Exercise training is the cornerstone of Pulmonary Rehabilitation (PR) inducing clinically meaningful improvements in exercise tolerance in patients with chronic obstructive pulmonary disease (COPD) [1]. Dynamic hyperinflation (DH) during exercise limits the normal increase in tidal volume, worsening breathlessness and reducing exercise capacity [2]. Additionally, DH and the concomitant high mean intrathoracic pressure swings are associated with adverse effects on central hemodynamic regulation, reducing the supply of oxygenated blood to deconditioned peripheral muscles [3,4]. This contributes to leg discomfort and further limits exercise tolerance. Different ergogenic strategies have been successfully employed to reduce exercise-induced breathlessness and leg discomfort, including oxygen and heliox supplementation, non-invasive ventilation (NIV) and various intermittent exercise modalities [5,6,7,8]. Oxygen supplementation is commonly used during exercise in order to reduce desaturation and breathlessness [9]. Heliox supplementation is beneficial but impractical and expensive; therefore, it has primarily been used for research purposes [10]. Standard NIV is infrequently used due to the application of bulky equipment and the need for close supervision during exercise training [11].

The VitaBreath (Philips, Respironics, Morrisville, PA, USA) is a portable, handheld, battery-powered, non-invasive ventilation device (pNIV) intended to reduce activity-related shortness of breath [12]. It delivers 18 cm H_2_O inspiratory and 8 cm H_2_O expiratory pressures, but can only be used during recovery periods interspersing bouts of physical activity. The purpose of this study was to test the effect of pNIV compared to pursed lip breathing (PLB) on exercise tolerance comprising two different intermittent modalities, namely moderate-intensity (6 min at 60% peak work rate (WRpeak)) and high-intensity (2 min at 80% WRpeak) exercise. As both modalities are recommended by the British Thoracic Society and joint American Thoracic Society/European Respiratory Society guidelines for PR [13,14], we wished to explore whether pNIV support was equally effective during exercise comprising different intensity and duration characteristics.

We hypothesised that the use of pNIV compared to PLB would increase exercise tolerance by reducing the intensity of breathlessness during bouts of moderate- or high-intensity intermittent exercise and would confer greater benefit if used more frequently with the high-intensity protocol. Besides the physiological implications of pNIV during controlled laboratory exercise conditions, we were interested in the effects of pNIV on anxiety, breathlessness, symptom burden and ability to perform daily life physical tasks. Accordingly, we assessed the frequency and ease of use of the VitaBreath device alongside symptom burden, as well as attitudes toward the device over 12 weeks following the completion of the exercise protocols.

## 2. Experimental Section

### 2.1. Study Population

Twenty-four patients with stable COPD who met the following criteria participated in the study: (1) male or female aged 40 years or older, (2) current or previous smoking history: 10 or more pack years, (3) spirometry confirmed stable COPD (GOLD stages II–IV) under optimal medical therapy and (4) substantial exercise-induced DH (i.e.,: change in inspiratory capacity from baseline >0.15 L or >4.5% of predicted resting inspiratory capacity (IC)) [2]. Exclusion criteria (Appendix B) included COPD exacerbation within 6 weeks, unstable comorbidities, inability to exercise and intolerance of the device. The investigations were carried out following the rules of the Declaration of Helsinki of 1975 [15], revised in 2013. NHS Research Ethics Committee approval (REC: 17/NE/0085) and Clinical Trials registration (NCT03068026) were obtained. Data were collected at the Respiratory Department, North Tyneside General Hospital. All participants provided written consent.

### 2.2. Study Design

This was a randomised, open-label cross-over trial comparing the use of pNIV to PLB during recovery periods in two different intermittent exercise regimes (Figure 1). Patients underwent a ramp incremental cardiopulmonary exercise test (CPET) to determine WRpeak, and then were randomly assigned to a high-intensity (HI) or a moderate-intensity (MOD) protocol. Within these groups, each patient performed two more visits using both pNIV and PLB during recovery from exercise in balanced order (see below); the primary outcome was exercise endurance time (TLim). Patients were on optimal bronchodilator therapy including daily LAMA and LABA and no changes to medication were made during the trial. Tests were performed without supplemental oxygen. Following the last visit, all 24 patients were given a VitaBreath device to use it as they wished and were contacted at 2 and 12 weeks to assess their use of, and attitudes towards, the device. Patients received advice on the use of the device for symptomatic relief after exertion, but the frequency of use was not prescribed. The full details of study methods are shown in Appendix B.

### 2.3. Assessments

Prior to the first exercise, test patients performed baseline screening comprising: (1) Spirometry, lung volume measurements and diffusion capacity (DLCO), (2) a resting electrocardiograph (ECG) and (3) medical history and physical examination by a clinician. In addition, they were familiarised with both high- and moderate-intensity exercise protocols. They practiced using the VitaBreath device and the correct adoption of PLB with guidance from a physiotherapist over 6–8 practice exercise sessions.

All CPET were performed on an electromagnetically braked cycle ergometer (Ergoselect 200, ergoline GmbH, Bitz, Germany) with the patients maintaining a pedalling frequency of 50–60 rpm. Pulmonary gas exchange and ventilatory variables were recorded breath-by-breath via a portable gas exchange analyser (K4b^2^, Cosmed, Shepperton, UK) throughout the test. The modified Borg Scale was used to rate the magnitude of breathlessness and leg discomfort at the end of the test [16]. Inspiratory capacity (IC) manoeuvres were performed at rest, every 2 min during cycling and at peak exercise, in order to evaluate the rate of dynamic hyperinflation (DH) [2]. Full details of CPET are shown in Appendix B.

### 2.4. Interventions

Patients randomised to moderate- or high-intensity intermittent exercise (Figure 1) underwent two tests within the allocated exercise protocol using either pNIV or PLB during recovery periods in a balanced order sequence (Figure 2). The order of the recovery method was determined by an alternating sequence, ensuring balance across the group. Throughout the test, pulmonary gas exchange variables were recorded breath-by-breath (K4b^2^, Cosmed, Shepperton, UK). Percentage arterial oxygen saturation (SpO_2_) was measured by a pulse oximeter every minute.

The high-intensity intermittent exercise protocol (sustained at 80% WRpeak) consisted of repetitive 2-min exercise bouts, separated by 2-min recovery periods to the limit of tolerance. This was defined as the point at which the patient signalled the inability to continue exercising or could not maintain the required pedalling rate (i.e., 50–60 revolutions/min) despite being encouraged by the investigators. IC manoeuvres were performed on the 2nd minute of each exercise bout. During the 1st min of each recovery period, participants used either pNIV (VitaBreath device) or PLB. During the 2nd min of each recovery period, participants performed an IC manoeuvre to assess the magnitude of DH and scored the intensity of their breathlessness and leg discomfort using the Borg scale (Figure 2a). SpO_2_ measurements were performed on the 2nd minute of each exercise bout following each IC manoeuvre.

The moderate-intensity intermittent exercise protocol (sustained at 60% WRpeak) consisted of repetitive 6-min exercise bouts, separated by 2-min recovery periods to the limit of tolerance as described above. Use of pNIV or PLB, and assessments were performed as above. IC manoeuvres, followed by SpO_2_ measurements, were performed on the 2nd, 4th and 6th minute of each exercise bout and on the 2nd minute of each recovery period following completion of each exercise bout (Figure 2b).

Cardiac Output (CO), Heart Rate (HR) and Stroke Volume (SV) were recorded continuously at rest, during exercise and in recovery by the Physio Flow device (Enduro, PF-07, Manatec Biomedical, Folschviller, France), whereas systemic oxygen delivery (DO_2_) was estimated from CO and SpO_2_ (Appendix B) [17,18,19,20,21].

Following completion of the three visits, all patients were provided with a VitaBreath device to use at home as they wished. Use of, and perceived benefit from, the VitaBreath device was examined at 2 and 12 weeks following the completion of the exercise testing protocols. The survey included questions on symptom burden, ability to perform daily tasks and perceived benefit from the device (Appendix A).

### 2.5. Statistics

Verification of sample size within each exercise modality was based on the study by Bianchi and colleagues [22] comparing pressure support ventilation (PSV) to control breathing during exercise. Using the mean difference in endurance time (3.4 min) between PSV and control breathing, and standard deviation (SD) (4.6 min), an alpha significance level of 0.05 (2-sided) and 90% power, a minimum total sample size of 10 patients was calculated to be sufficient to detect significant differences in endurance time between pNIV and PLB trials within each exercise modality (Appendix B). Randomisation was performed by independent staff within our institution and stratified by WRpeak (<50 or ≥50 watts) and FEV_1_ (<50 or ≥50% predicted) using a block size of 4. The study team was blinded to the randomisation sequence.

Data are presented as mean ± SD unless otherwise stated. One-way ANOVA was employed to detect differences in exercise tolerance (minutes) between moderate- and high-intensity exercise modalities and between pNIV and PLB breathing modalities. For each individual patient, the duration of exercise to the limit of tolerance when using the PLB technique was divided into four percentiles (i.e., 25%, 50%, 75% and 100%) of total endurance time including the recovery phases. A two-way ANOVA with repeated measurements followed by appropriate post hoc analysis was employed to compare changes at iso-time across these four percentiles between the PLB and pNIV trials for: IC, CO, DO_2_, breathlessness and leg discomfort. For IC we calculated the change between recovery periods and the end of exercise bouts. When analysing the questionnaire results, data are presented as median (IQR) or absolute number (%). The Wilcoxon signed-rank test was used for comparing 2- and 12-week interval scale (Likert style) data and McNemar’s test for nominal response data. The level of significance for all analyses was set at *p* < 0.05.

## 3. Results

### 3.1. Study Population

Patients randomised (between June 2017 and July 2018) to exercise modalities were matched in terms of demographic and clinical characteristics exhibiting severe airflow limitation and lung hyperinflation at rest without resting hypoxemia (Table 1). Baseline peak exercise capacity was severely impaired; patients exhibited profound exercise-induced DH and moderate arterial oxygen desaturation at the limit of tolerance during the CPET (Table 2). Five (20.8%) of the patients were current smokers, with 14 (58.3%) having been admitted to hospital for an exacerbation of COPD (ECOPD) in the past 12 months. The median number of ECOPD (hospital or community managed) in the past 12 months was 2 (1–4.75). The median extended MRC dyspnoea score (eMRCD) was 4 (3–4) [23,24].

### 3.2. Endurance Time

Compared to PLB, pNIV increased exercise endurance time during for both intermittent exercise modalities (HI-pNIV: from 26.2 ± 6.9 to 31.4 ± 8.3 min (*p* = 0.008); MOD-pNIV: from 30.3 ± 11.3 to 36.1 ± 11.0 min (*p* = 0.016)) without differences in the magnitude of improvement between the two exercise modalities (*p* = 0.244) (Figure 3).

### 3.3. Other Outcomes

The mean increase in IC during recovery periods compared to the end of exercise exceeded the clinically meaningful margin (i.e., >4.5% of predicted resting IC: 110–119 mL) [2,7] when patients used pNIV for both HI-pNIV: 140 ± 110 mL and MOD-pNIV: 170 ± 80 mL exercise modalities (Figure 4a,c). The mean change in IC in recovery compared to the end of exercise with PLB did not reach a clinically meaningful margin neither for HI-PLB: 10 ± 290 mL nor for MOD-PLB: 100 ± 140 mL exercise modalities (Figure 4a,c).

Compared to PLB at Tlim, the change in IC from rest was not different (*p* = 0.379) between the two intermittent exercise modalities with the use of pNIV (Table 3).

Compared to PLB across different fractions of total endurance, time application of pNIV was associated with a clinically meaningful reduction [7,25] in breathlessness during HI-pNIV (by: 1.2 ± 1.7, *p* = 0.022) and MOD-pMIV (by: 1.0 ± 0.7, *p* = 0.002) exercise modalities (Figure 4b,d). There were no significant differences (*p* = 0.518) in breathlessness scores between the two modalities with pNIV.

Compared to PLB across different fractions of total endurance time, the application of pNIV was associated with lower leg discomfort during HI-pNIV (by: 0.5 ± 0.8, *p* = 0.050) and MOD-pNIV (by: 0.8 ± 1.1 (*p* = 0.031)) exercise modalities (Figure 5c,f). There were no differences (*p* = 0.268) in leg discomfort scores between the two modalities with pNIV.

In comparison to PLB, across different fractions of total endurance time, CO and DO_2_ were greater with pNIV during both HI-pNIV (by 0.3 ± 1.1 L/min (*p* = 0.035) and by 70 ± 40 mL/min (*p* = 0.040), respectively) and MOD-pNIV (by 0.8 ± 0.9 L/min (*p* = 0.045) and 160 ± 40 mL/min (*p* = 0.040), respectively) exercise modalities (Figure 5). There were no differences (*p* = 0.519 and 0.463, respectively) in these variables between the two exercise modalities with pNIV (Figure 5).

### 3.4. Use and Perceived Benefits of the VitaBreath Device

Compared to the pre-VitaBreath period, at 12 weeks, patients were significantly less anxious about becoming breathless on a 10-point Likert Scale: median (IQR) pre-VitaBreath = 7.5 (5.25–8.75); 12 weeks = 4.0 (2–5.75); (*p* = 0.001) and 17 of 24 patients perceived a shorter time to recovery from breathlessness (*p* = 0.002) (Table 4).

Comparing responses at 2 and 12 weeks, there was no attrition in the frequency of use of the device (*p* = 0.590); at 12 weeks, 23/24 patients continued to use the device at least weekly, and at 16 weeks, daily. Patients found improvement in the speed, duration and confidence with which they could undertake activities of daily living, with no loss of these effects at 12 weeks (Appendix A). Frequency of breathlessness (*p* = 0.670), planning around breathlessness (*p* = 0.220), and needing to stop activities due to breathlessness (*p* = 0.500) did not change between 2 and 12 weeks. Whilst all patients describe the VitaBreath as easy to use (good or better), most describe its portability as poor (8/24) or fair (7/24). Patients described being more active with VitaBreath than without (2 weeks 13/23; 12 weeks 14/24 patients; *p* = 0.940) (Appendix A). Between 2 and 12 weeks, there was a non-significant increase in use of VitaBreath for outdoor activities (15/24 vs. 19/23 patients; *p* = 0.130). At 12 weeks, the median (IQR) likelihood that patients would recommend VitaBreath to others = 10 (7–10), but to purchase at the actual cost in sterling pounds = 3 (1–5) on a 10-point likert scale (Appendix A).

## 4. Discussion

The major finding of the study is that the use of pNIV during recovery periods interspersing moderate- and high-intensity bouts of intermittent exercise significantly improved exercise tolerance compared to PLB. This is probably due to more rapid recovery from exercise-induced dynamic hyperinflation, with associated improvements in cardiac output and systemic oxygen delivery. The physiological responses shown were matched by a reduction in breathlessness and leg discomfort in recovery from exercise. Patients reported that the VitaBreath device improved anxiety around breathlessness, as well as perceived time of recovery from it during activities of daily living.

NIV has previously been used during exercise training in patients with severe COPD to lessen breathlessness and increase exercise capacity [11,26]. A Cochrane analysis of studies using NIV during exercise training provides conflicting and moderate quality evidence of beneficial effects on exercise capacity and the role of this intervention remains unclear [27,28,29,30,31]. The main drawbacks of NIV in these studies were the difficulty in using the equipment during pulmonary rehabilitation and cost, including the time required to supervise patients during training [29]. Nevertheless, recent meta-analyses conclude that there is a need for further randomised clinical trials [29,32].

The VitaBreath device is designed to overcome the problems associated with the use of traditional NIV and is primarily intended to aid recovery from breathlessness after activities in daily life. It is light, handheld and battery operated. This, and future pNIV devices, may offer benefits within PR, particularly in intermittent/interval training regimes. One technical limitation of the VitaBreath device is that the expiratory and inspiratory positive airway pressures (EPAP and IPAP, respectively) are fixed. Excessive EPAP can worsen hyperinflation and circulatory compromise. In our study population, the majority of patients showed no worsening of DH or circulatory compromise whilst using pNIV during recovery periods. However, in six patients (three per group) the improvement in DH was greater with PLB than pNIV, thereby suggesting that the fixed pressures were sub-optimal in at least some of our cohort. Accordingly, in future devices the ability to adjust EPAP and pressure support is desirable and could potentially be automated.

Recently, a non-invasive “open” ventilation (NIOV) system operating in conjunction with a portable oxygen tank was found to decrease respiratory muscle activation and dyspnoea and to improve cycle ergometer exercise tolerance [33]. We have shown that the use of pNIV (VitaBreath) during intermittent exercise is associated with longer exercise endurance time (by 19–20%), with less DH and breathlessness. Continuous positive pressure support throughout exercise is much less practical in daily life, but confers greater improvement in exercise tolerance, including the use of: continuous positive airway pressure (26–40%) [22,34], proportional assist ventilation (23–43%) [17,22,34], pressure support ventilation (by 32–38%) [22,35], inspiratory pressure support (46%) [36] and NIOV (by 245%) [33]. Notably, when the NIOV system was powered by compressed air rather than oxygen, exercise endurance time was increased only by 13% [33]. The aforementioned ventilation support strategies provide continuous unloading of the respiratory muscles and reduce the work of breathing [11,26], and as such, are expected to yield greater improvement in exercise capacity compared to the intermittent application of pNIV during recovery from exercise. This argument is further supported by the absence of a reduction in ventilatory requirement or an increase in arterial oxygen saturation during the successive bouts of high- or moderate-intensity exercise with the application of pNIV compared to PLB in the present study (Table 3). Whilst intermittent use of pNIV offered less improvement in exercise tolerance, it is more practical in typical PR settings and daily life. Studies in the future may investigate the additive effect of oxygen supplementation to intermittent NIV support during typical PR.

Our findings suggest that pNIV improved exercise tolerance compared to PLB not just by its direct effect on respiratory mechanics (reducing DH), but also by partial alleviation of the associated adverse hemodynamic responses. Use of pNIV resulted in an increase in cardiac output and systemic oxygen delivery during exercise. This is most likely as a result of reduced DH and improved venous return, which is in line with previously published reports following application of PAV [17] and administration of bronchodilators or heliox [37,38]. This finding confirms that a common basis for enhanced exercise performance in COPD may be associated with improved peripheral muscle oxygen availability and reduced symptoms of leg discomfort afforded by interventions targeting the abnormal respiratory mechanics in COPD.

VitaBreath provides positive inspiratory pressure support to reduce the work of breathing and positive expiratory pressure to keep the airways open during expiration, thereby reducing air trapping [12]. The mean increase in IC during the recovery periods compared to the end of exercise bouts with pNIV (high-intensity: 140 mL; moderate-intensity: 170 mL) was within the clinically meaningful margin for bronchodilator trials (138–175 mL) [39], most likely reflecting the improvement in expiratory flow and thus lung emptying [7,39,40]. The increase in IC during recovery was matched by a significant reduction in breathlessness that reached the minimal clinically important difference (1.0 unit) [7,40]. In contrast, PLB was not consistently associated with a clinically meaningful improvement in IC during recovery; this is in keeping with previous work showing that exercise-induced DH normally persists for several minutes following the end of exercise [41]. Furthermore, the mean reduction in breathlessness scores during recovery from exercise did not reach the minimal clinically important difference [7,40].

We used two intermittent protocols with different duration and intensity characteristics to explore the influence of intermittent pNIV on exercise tolerance in COPD patients with baseline and exercise-induced dynamic hyperinflation. We expected that the use of pNIV during recovery following successive 2-min exercise bouts at high-intensity would have conferred greater benefit being used more frequently compared to 6-min bouts of moderate-intensity exercise. Our hypothesis was not confirmed. The high-intensity protocol elicited greater exercise-induced DH compared to the moderate intensity protocol (Table 3), hence the effectiveness of positive pressure ventilation on exercise tolerance, physiological responses and symptoms is highly comparable between the two intermittent modalities. Our findings suggest that pNIV could be applied in the PR setting to either prolong endurance at a given training load sustained at moderate intensity and/or to allow greater training loads such as the high-intensity protocol in the present study. Further research is required to confirm this as our study provides evidence only during acute application of portable NIV; hence these data cannot be extrapolated to give information about the effects during a long period of training.

All participants were given a device to use in daily life. Our data suggests that the VitaBreath device improved anxiety around breathlessness, as well as perceived time of recovery from it. These findings further support the clinically meaningful improvement of inspiratory capacity and breathlessness during recovery following the termination of high- and moderate-intensity exercise. Patients felt benefits in speed, duration and confidence associated with their activities of daily living, which were maintained at 3 months. The VitaBreath device does not change the underlying disease process and, as expected, there was no change in frequency of breathlessness, nor the need to plan around, or stop activities due to breathlessness. Patients describe the device as easy-to-use, but its portability is unfavourable. However, it is important to note that over 95% of patients continue to use the device regularly, and most would recommend it to another person with COPD. Despite these benefits, few patients would purchase a device. This may, at least in part, reflect the UK National Health Service, which provides treatment free at the point of delivery, and the socioeconomic status of this patient group.

This is a single centre study and blinding the patients or investigators to the breathing modality was not possible due to the lack of a sham pNIV device. Thus, the risk of a placebo effect cannot be excluded, especially when considering the effect of the pNIV device on the reduction in breathlessness. We were unable to measure the work of breathing directly during the recovery periods from exercise. This would have allowed us to assess the effect of pNIV on respiratory muscle unloading. Use of optoelectronic plethysmography would have allowed continuous assessment of end-expiratory volumes in the transition from exercise to recovery [41]. Although most patients continued to use the device in daily life, with no attrition in reported benefits over 12 weeks, these results may have been influenced by study participation, including experience using the device under supervision during recovery from moderate or intense exercise.

## 5. Conclusions

Use of pNIV during the recovery periods interspersing moderate or high intensity exercise bouts enhances exercise tolerance compared to pursed lip breathing by lessening the symptoms of breathlessness and by enhancing systemic oxygen availability. Future studies may investigate the applicability and benefits of intermittent application of other available positive pressure ventilation strategies during the recovery periods of high-intensity intermittent exercise in COPD.

## Figures and Tables

**Figure 1 jcm-08-00094-f001:**
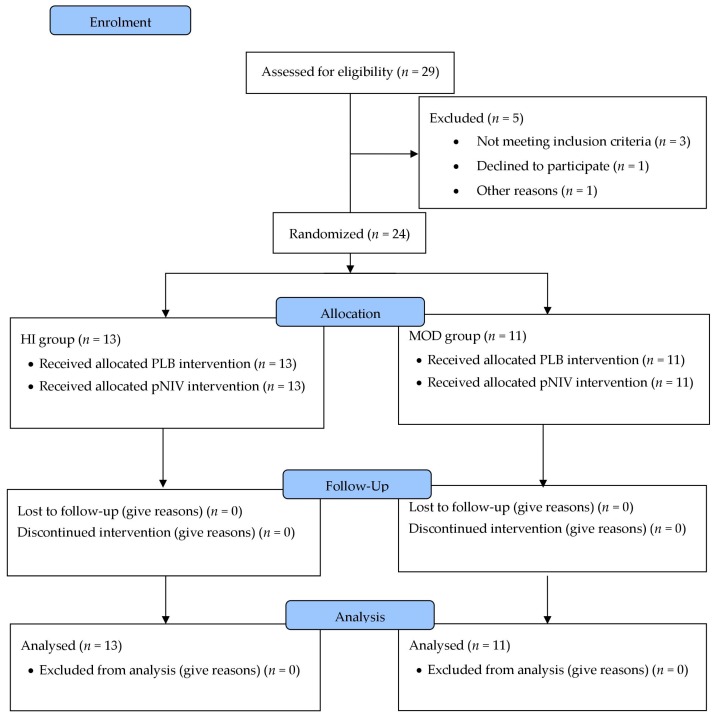
Consort diagram. Following the initial ramp incremental cardiopulmonary exercise test (CPET) to determine WRpeak, 24 patients were randomly allocated either to a high-intensity (HI) or a moderate-intensity (MOD) exercise protocol. Within these two protocols (HI or MOD), each patient performed two more exercise tests using both portable non-invasive ventilation (pNIV) and pursed lip breathing (PLB) during recovery from exercise in balanced order, alternating which test was performed first.

**Figure 2 jcm-08-00094-f002:**
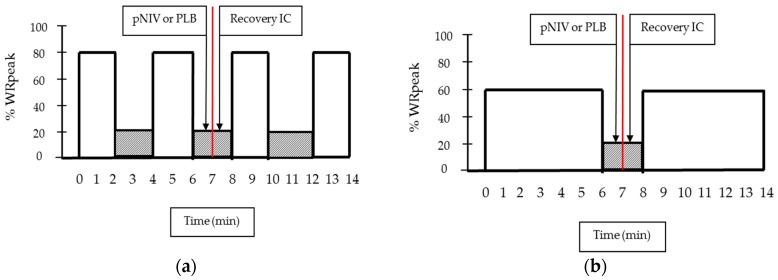
Exercise protocols: (**a**) High-intensity 2-min intermittent exercise protocol and (**b**) moderate-intensity 6-min intermittent exercise protocol. Within these two protocols (HI or MOD), each patient performed two more visits/exercise tests using pNIV and PLB during recovery from exercise in balanced order. Typical examples of high- and moderate-intensity exercise bouts are shown by open squares and recovery periods by shadowed squares.

**Figure 3 jcm-08-00094-f003:**
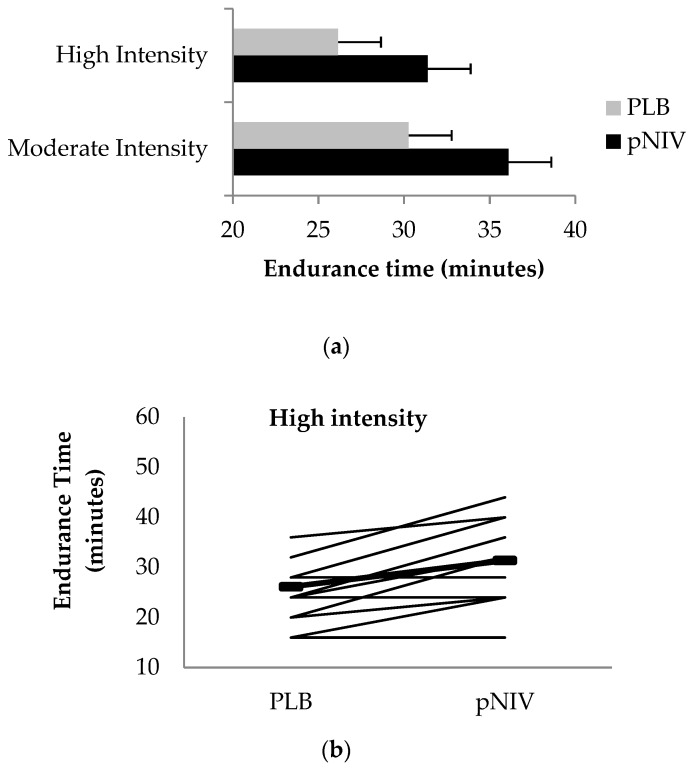
Exercise endurance time using pNIV (solid bars) and PLB (grey bars) during high- and moderate-intensity exercise protocols (**a**). Effect of pNIV on individual patient exercise endurance time during high-intensity (**b**) and moderate-intensity (**c**) exercise protocols. Solid lines indicate mean values.

**Figure 4 jcm-08-00094-f004:**
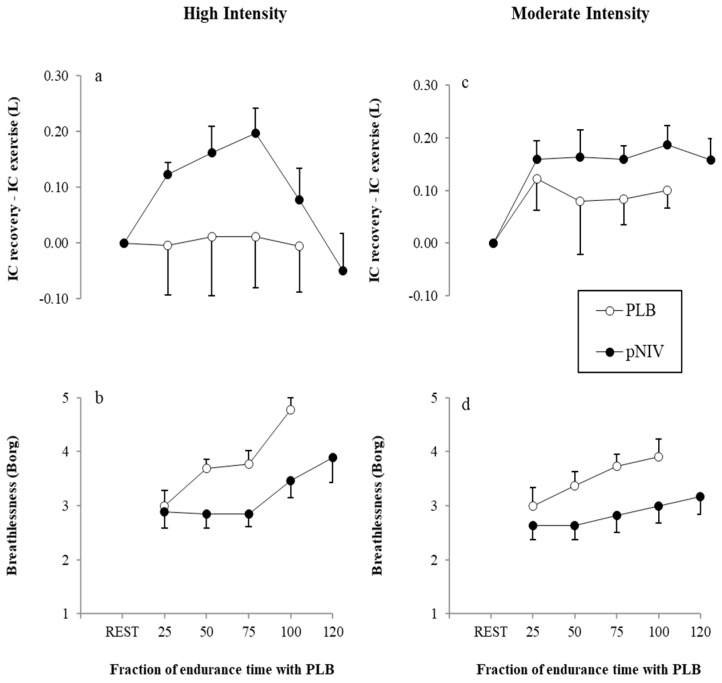
Effect of the application of pNIV (closed circles) compared to PLB (open circles) on inspiratory capacity calculated as the change between recovery periods and the end of high-intensity intermittent (**a**) or moderate-intensity (**b**) and moderate-intensity (**c**) exercise bouts and symptoms of breathlessness during recovery from high-intensity (**d**) exercise. Responses are shown for both PLB and pNIV at iso-time across the four percentiles (25%, 50%, 75% and 100%) of the total endurance time when using the PLB technique. Data are presented as mean ± standard error of the mean (SEM).

**Figure 5 jcm-08-00094-f005:**
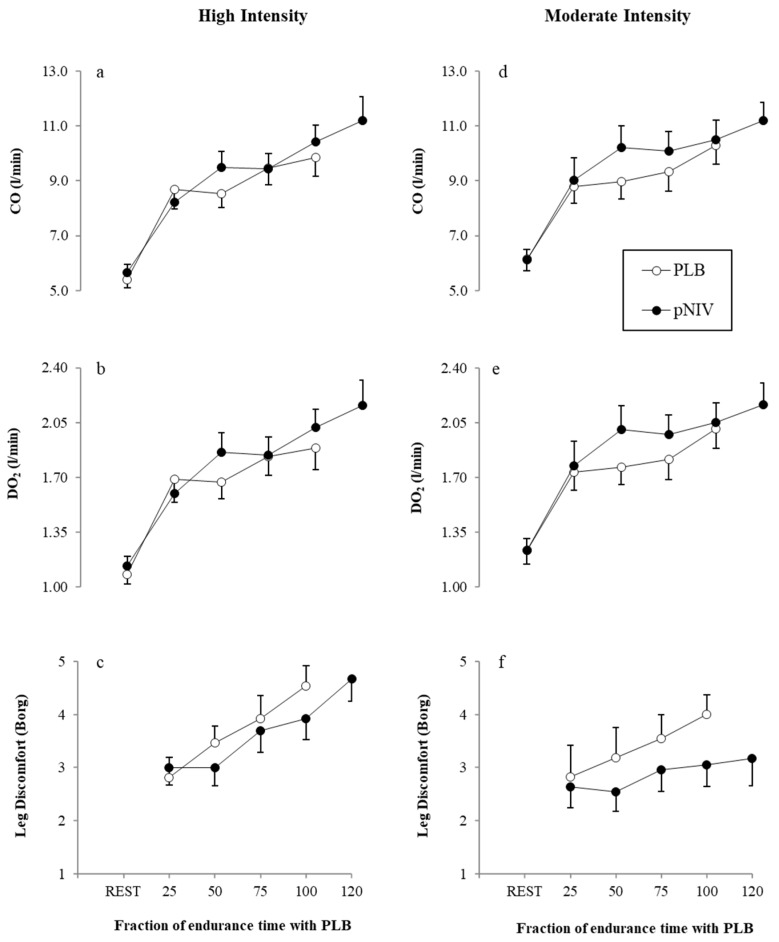
Effect of the application of pNIV (closed circles) compared to PLB (open circles) on cardiac output (CO: **a**,**d**), estimated systemic oxygen delivery (DO_2_: **b**,**e**) and symptoms of leg discomfort (**c**,**f**) during high- and moderate-intensity exercise protocols, respectively. Responses are shown for both PLB and pNIV at iso-time across the four percentiles (25%, 50%, 75% and 100%) of the total endurance time when using the PLB technique. Data are presented as mean ± SEM.

**Table 1 jcm-08-00094-t001:** Patient demographic characteristics at baseline.

Variable	High-Intensity (*n* = 13)	Moderate-Intensity (*n* = 11)	*p*-Value
Gender (M/F)	5/8	5/6	
Age (years)	66 ± 7	68 ± 10	0.510
BMI	26.9 ± 6.9	25.6 ± 6.8	0.659
FEV_1_ (% predicted)	46 ± 15	46 ± 21	0.948
FVC (% predicted)	87 ± 18	91 ± 21	0.605
FEV1/FVC (%)	43 ± 14	37 ± 12	0.487
IC (litres)	1.96 ± 0.56	2.03 ± 0.78	0.810
IC (% predicted)	79 ± 22	78 ± 23	0.807
TLC (% predicted)	130 ± 29	131 ± 15	0.975
FRC (% predicted)	172 ± 49	175 ± 37	0.845
DLCO (% predicted)	38 ± 18	38 ± 20	0.980

BMI, body mass index; FEV_1_, forced expiratory volume in the first second; FVC, forced vital capacity; IC, inspiratory capacity; TLC, total lung capacity; FRC, functional residual capacity; DLCO, transfer factor of the lung for carbon monoxide; M, male; F, female; values are mean ± standard deviation (SD).

**Table 2 jcm-08-00094-t002:** Peak physiological variables at the limit of tolerance during cardiopulmonary exercise test (CPET).

Variable	High-Intensity (*n* = 13)	Moderate-Intensity (*n* = 11)	*p*-Value
WR (Watts)	48 ± 25	48 ± 26	0.977
WR (% predicted)	46 ± 19	45 ± 26	0.883
VO_2_ (mL/kg/min)	13.5 ± 3.9	13.4 ± 3.2	0.808
VO_2_ (% predicted)	60 ± 12	61 ± 21	0.911
VE/MVV (%)	1.00 ± 0.21	0.99 ± 0.26	0.787
ΔIC from rest (litres)	0.60 ± 0.38	0.47 ± 0.33	0.399
SpO_2_ (%)	92 ± 5	92 ± 3	0.827
CO (L/min)	10.5 ± 3.9	11.2 ± 2.7	0.635
HR (beats/min)	113 ± 15	110 ± 18	0.728
SV (mL)	94 ± 30	101 ± 19	0.513
Dyspnoea (Borg 1–10)	4.2 ± 1.2	3.8 ± 0.7	0.269
Leg discomfort (Borg 1–10)	3.8 ± 1.1	3.3 ± 1.6	0.442

WR, work rate; VO_2_, oxygen uptake; VE, minute ventilation; MVV, Maximum voluntary ventilation; ΔIC, change from rest in inspiratory capacity; SpO_2_, arterial oxygen saturation; CO, cardiac output; HR, heart rate; SV, stroke volume; Values are mean ± SD.

**Table 3 jcm-08-00094-t003:** Metabolic and respiratory responses at the limit of tolerance of high- and moderate-intensity exercise.

	High-Intensity			Moderate-Intensity		
Variable	PLB	pNIV Support	*p*-Value	PLB	pNIV Support	*p*-Value
Work Rate (watts)	38 ± 20	38 ± 20	-	30 ± 17	30 ± 17	-
ΔIC (litres)	−0.37 ± 0.31	−0.37 ± 0.28	0.964	−0.29 ± 0.25	−0.27 ± 0,24	0.819
Dyspnoea (Borg)	4.8 ± 1.2	3.9 ± 1.4	0.005	4.0 ± 1.1	3.3 ± 1.1	0.004
Leg Discomfort (Borg)	4.5 ± 1.5	4.1 ± 1.8	0.027	4.1 ± 1.2	3.3 ± 1.6	0.011
VO_2_ (mL/min/kg)	11.89 ± 3.20	11.95 ± 3.45	0.879	12.82 ± 3.27	12.72 ± 3.11	0.778
VE (litres/min)	34.75 ± 18.43	36.47 ± 18.12	0.206	34.57 ± 14.64	33.84 ± 14.25	0.444
VT (litres)	1.2 ± 0.5	1.2 ± 0.5	0.202	1.2 ± 0.5	1.2 ± 0.5	0.549
RF (breaths/min)	29 ± 5	30 ± 5	0.256	30 ± 5	29 ± 3	0.409
CO (litres/min)	9.9 ± 2.6	10.5 ± 2.7	0.070	10.3 ± 2.4	10.5 ± 2.3	0.193
HR (beats/min)	105 ± 13	108 ± 15	0.172	114 ± 19	114 ± 16	0.725
SV (mL/beat)	94 ± 23	96 ± 22	0.487	90 ± 16	91 ± 13	0.485
SBP (mmHg)	146 ± 23	158 ± 23	0.002	142 ± 29	148 ± 22	0.285
DBP (mmHg)	82 ± 8	89 ± 9	0.024	78 ± 18	84 ± 17	0.049
a-VO_2_ (mL/100 mL)	8.9 ± 2.9	8.3 ± 3.1	0.205	8.9 ± 2.9	8.6 ± 2.9	0.371
SpO_2_ (%)	92 ± 5	93 ± 4	0.104	94 ± 3	93 ± 3	0.148

ΔIC: change from baseline in inspiratory capacity, VO_2_: oxygen uptake, VE: minute ventilation, VT: tidal volume, RF: breathing frequency, CO: cardiac output, HR: heart rate, SV: stroke volume, SBP: systolic blood pressure, DBP: diastolic blood pressure, a-VO_2_: whole body arteriovenous oxygen difference content, SpO_2_: arterial oxygen saturation. pNIV: portable Non-Invasive Ventilation, PLB: Pursed lip breathing technique. Values are mean ± SD.

**Table 4 jcm-08-00094-t004:** Effects of the use of VitaBreath device on anxiety, symptom burden and ability to perform tasks.

Question	Pre-VitaBreath	Post-VitaBreath	*p*-Value
How anxious are you about becoming short of breath (SOB)? 1 = Not at all anxious 10 = Very anxious	7.50 (5.25–8.75)	4.00 (2–5.75)	0.001 * 20 improvements 2 worse 2 ties
How long did it take you to recover from SOB?			0.002 * 17 improvements 3 worse 4 ties
<1 min	0 (0%)	6 (25%)	
2–3 min	7 (29.2%)	9 (37.5%)	
4–5 min	3 (12.5%)	4 (16.7%)	
5–7 min	5 (20.8%)	1 (4.2%)	
7–10 min	5 (20.8%)	2 (8.3%)	
More than 10 min	4 (16.7%)	2 (8.3%)	

Data presented as median (IQR) or absolute number (%); * Wilcoxon signed-rank test.

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
