# Peer review of "Intermittent Use of Portable NIV Increases Exercise Tolerance in COPD: A Randomised, Cross-Over Trial"

_jcm, 2019, doi:10.3390/jcm8010094_

Reviewer 1 Report

I congratulate for the study proposed to the Journal.

The study appears complex, intriguing and offers important information, especially physiological one.

 I ask the authors to improve the usability and clarity of the paper on the following points:

1. Figure 1 is not clear; it is not clear where the cross-matched randomization between pNIV and PLB is done

2. It is necessary to clarify better that the HI and MOD sessions proposed to the patient were consecutive and repetitive until the patient is exhausted

3. the criteria for patient exhaustion must be clarified

4. it is necessary to clarify what was done at visit 2 with respect to visit 3

5. It is necessary to clarify what it means that patients could use the pNIV instrument at their own decision at home without a real proposed FITT protocol

6. Have all 24 patients been given the instrument?

7. It is not completely clear to me how the comparison data between pNIV and PLB were made at  isotime

8. Figure 2 (part a and part b) is unclear: here it is also necessary to distinguish what was done at visit 2 and 3, where there was randomization between pNIV and PLB and  the time to exhaustion should be represented with a punctuated line.

9. in the figures 2a and 2b the words “Recovery IC” are not clear

10. perhaps the authors could use for clarity the achronimous of the 4 tests: Mod-pNIV, Mod-PLB, HI-pNIV and HI-PLB

11. The interventions section of page 5 should be better re-arranged (with the help of a new clearer figure)

12. an explanatory figure of the main outcome (endurance time) with individual and average data would be very informative

13. Figure 3 and Figure 4: the legend with "Responses are shown ......... .was adopted" is obscure

14. All tests were performed without oxygen supplement obviously: all patients were normoxemic?

15. A protocol that had compared the 4 tests on the same patient rather than comparing two different groups would have been more informative

16. The authors do not discuss the problem of the fixed setting of the device (18 cmh20 and 8 cmH20) with possible risks of over-assistance, under assistance or mismatch in particular for the fixed ext-PEEP that could cause a worsening of the intrinsic-PEEP

17. The study is a physiological study during an acute effect study; data can not be extrapolated to give information about the effects during a long period of training: please stress this point

18. What results would the authors expect if oxygen therapy was added to the pNIV or PLB?

19. The risk of a placebo effect can not be excluded, especially for the reduction in dyspnoea: this should be better stressed even if the authors have already mentioned the impossibility of a study with a sham device

Author Response

Reviewer 1:

Comment

I congratulate for the study proposed to the Journal.

The study appears complex, intriguing and offers important information, especially physiological one.

Response

We would like to thank the reviewer for his/her constructive comments and suggestions that have ultimately improved our manuscript. We have addressed all comments in the revised version of the manuscript. Revisions are shown as tracked changes.

Comment

I ask the authors to improve the usability and clarity of the paper on the following points:

Figure 1 is not clear; it is not clear where the cross-matched randomization between pNIV and PLB is done

Response

We have addressed the reviewer’s concern in the revised figure 1. Patients were randomized to one of the two exercise regimes, stratified by FEV1 and WRpeak. Within these randomised groups patients performed both pNIV and PLB. ‘The order of the recovery method was determined by an alternating sequence, ensuring balance across the group’ as it is now indicated on pages 4 and 5 of the revised version. In addition in the legend of figure 1 we indicate that: ‘Following the initial ramp incremental cardiopulmonary exercise test (CPET) to determine WRpeak, 24 patients were randomly allocated either to a high-intensity (HI) or a moderate-intensity (MOD) exercise protocol.  Within these two protocols (HI or MOD) each patient performed two more exercise tests using both pNIV and PLB during recovery from exercise in balanced order, alternating which test was performed first’.

Comment

2. It is necessary to clarify better that the HI and MOD sessions proposed to the patient were consecutive and repetitive until the patient is exhausted

Response

Indeed, in the revised version of the manuscript on p.5 (second and third paragraphs) we indicate for both HI and MOD exercise protocols that bouts were repetitive to the limit of tolerance.

Comment

3. the criteria for patient exhaustion must be clarified

Response

Following the reviewer’s enquiry we state on p.5 of the revised manuscript that: ‘during HI and MOD exercise protocols patients cycled to the limit of tolerance. This was defined as the point at which the patient signalled the inability to continue exercising or could not maintain the required pedalling rate (i.e. 50-60 revolutions/min) despite being encouraged by the investigators’.

Comment

4. it is necessary to clarify what was done at visit 2 with respect to visit 3

Response

In the revised manuscript on p. 2 under the section of study design we clarify that: ‘During visit one, patients underwent a ramp incremental cardiopulmonary exercise test (CPET) to determine WRpeak, and then were randomly assigned to a high-intensity (HI) or a moderate-intensity (MOD) protocol, and within these groups each patient performed two more visits using pNIV and PLB during recovery from exercise in balanced order’.

Comment

5. It is necessary to clarify what it means that patients could use the pNIV instrument at their own decision at home without a real proposed FITT protocol

Response

In the revised version of the manuscript we state on p.2 under the section of study design the following: ‘Following the last visit all 24 patients were given a VitaBreath device to use it as they wished and contacted at 2 and 12 weeks to assess their use of, and attitudes toward, the device. Patients received advice on use of the device for symptomatic relief after exertion, but the frequency of use was not prescribed’.

Comment

6. Have all 24 patients been given the instrument?

Response

In the revised manuscript on p. 2 under the section of study design we clarify that: ‘Following the last visit all 24 patients were given a VitaBreath device’.

Comment

7. It is not completely clear to me how the comparison data between pNIV and PLB were made at  isotime

Response

On p.5 under the section of statistics we clarify that: ‘For each individual patient the duration of exercise to the limit of tolerance when using the PLB technique was divided into four percentiles (i.e.: 25, 50, 75 and 100%) of total endurance time including the recovery phases. A two-way ANOVA with repeated measurements followed by appropriate post hoc analysis was employed to compare changes at iso-time across these four percentiles between the PLB and pNIV trials for: IC, CO, DO2, breathlessness and leg discomfort.’ This is also clarified in the legends of figures 4 and 5 of the revised manuscript.

Comment

8. Figure 2 (part a and part b) is unclear: here it is also necessary to distinguish what was done at visit 2 and 3, where there was randomization between pNIV and PLB and the time to exhaustion should be represented with a punctuated line.

Response

As explained in the section of study design following the initial ramp incremental cardiopulmonary exercise test (CPET) to determine WRpeak, 24 patients were randomly allocated either to a high-intensity (HI) or a moderate-intensity (MOD) protocol.  Within these two protocols (HI or MOD) each patient performed two more visits/exercise tests using pNIV and PLB during recovery from exercise in balanced order (i.e. to avoid bias of using PLB in the first visit). This is now clear in the consort diagram legend (Figure 1). In terms of time to exhaustion within each exercise modality and breathing technique, we now provide a new figure (Figure 3) clearly showing the time of exhaustion (Figure 3a) across the two exercise modalities and breathing techniques. Thank you for your valuable suggestion.

Comment

9. in the figures 2a and 2b the words “Recovery IC” are not clear

Response

We have increased the size of the font in this figure. Thank you.

Comment

10. perhaps the authors could use for clarity the achronimous of the 4 tests: Mod-pNIV, Mod-PLB, HI-pNIV and HI-PLB

Response

In the section of results we have used the abbreviations suggested by the reviewer.

Comment

11. The interventions section of page 5 should be better re-arranged (with the help of a new clearer figure)

Response

Thank you for your suggestion. This section now contains more information and reference to amended figures 1 and 2.

Comment

12. an explanatory figure of the main outcome (endurance time) with individual and average data would be very informative

Response

We have produced a new figure (figure 3) displaying both individual and average data of endurance time for HI-pNIV, HI-PLB, MOD-pNIV and MOD-PLB as requested. Thank you for yours suggestion.

Comment

13. Figure 3 and Figure 4: the legend with "Responses are shown ......... .was adopted" is obscure

Response

As stated above we have modified the legends of both figures (now Figures 4 and 5) indicating that: ‘Responses are shown for both PLB and pNIV at iso-time across the four percentiles (25%, 50%, 75% and 100%) of the total endurance time when using the PLB technique’.

Comment

14. All tests were performed without oxygen supplement obviously: all patients were normoxemic?

Response

In the section of study design of the revised manuscript we now make clear that all tests were performed without supplemental oxygen (p.2), whereas in the section of results we declare that patients were normoxemic at rest (p.6).

Comment

15. A protocol that had compared the 4 tests on the same patient rather than comparing two different groups would have been more informative

Response

Certainly this would have been the ideal scenario, but as this reviewer is aware it is difficult to perform multiple experiments in this population.

Comment

16. The authors do not discuss the problem of the fixed setting of the device (18 cmh20 and 8 cmH20) with possible risks of over-assistance, under assistance or mismatch in particular for the fixed ext-PEEP that could cause a worsening of the intrinsic-PEEP

Response

We agree with the reviewer; this is a limitation we discussed with the manufacturer prior to the study; unfortunately it was not possible to allow modification of IPAP or EPAP. We have added the following comment to the discussion on p. 12: “One technical limitation of the VItaBreath device is that the expiratory and inspiratory positive airway pressures (EPAP and IPAP, respectively) are fixed. Excessive EPAP can worsen hyperinflation and circulatory compromise. In our study population the majority of patients showed no worsening of DH or circulatory compromise whilst using pNIV during recovery periods. However in six patients (three per group) the improvement in DH was greater with PLB than pNIV, thereby suggesting that the fixed pressures were sub-optimal in at least some of our cohort. Accordingly, in future devices the ability to adjust EPAP and pressure support is desirable and could potentially be automated. “

Comment

17. The study is a physiological study during an acute effect study; data can not be extrapolated to give information about the effects during a long period of training: please stress this point

Response

We agree with this reviewer and we have included the following statement in the section of discussion on p. 13: ‘Further research is required to confirm this as our study provides evidence only during acute application of portable NIV; hence these data cannot be extrapolated to give information about the effects during a long period of training’.

Comment

18. What results would the authors expect if oxygen therapy was added to the pNIV or PLB?

Response

This is a very nice point and something to investigate in the future in the setting of pulmonary rehabilitation. Accordingly, in the revised version of the manuscript on p. 12 we suggest the following: ‘Studies in the future may investigate the additive effect of oxygen supplementation to intermittent NIV support during typical PR’. 

Comment

19. The risk of a placebo effect can not be excluded, especially for the reduction in dyspnoea: this should be better stressed even if the authors have already mentioned the impossibility of a study with a sham device

Response

We appreciate the point made by the reviewer. Accordingly, in the section of study limitations of the revised manuscript on p. 13 we state the following: ‘Thus, the risk of a placebo effect cannot be excluded, especially when considering the effect of the pNIV device on the reduction in breathlessness’.  

Reviewer 2 Report

This is an interesting study proving the applicability of a portable NIV device to mitigate exercise intolerance in COPD. The article is well written, despite the seemingly low number of participants the differences seem to be robust.

Comments:

·      Patient characteristics. The authors should add drug therapy. Bronchodilators administered prior to exercise can reduce dynamic hyperinflation themselves. Were they encouraged/restricted to take their medication prior to exercise? Please, discuss.

·       NIV. Why IPAP of 18 and EPAP of 8 cmH2O? Can these settings be modified? Pressures affect dynamic hyperinflation and I guess not all patients require the same settings. Please, discuss.

·       The authors may consider removing Figure 2C as it does not add to the manuscript but may serve commercial benefit.

·       Table 1 and 2. Please provide p values for comparison.

·       Table 1. Please add units for the parameters

Author Response

Reviewer 2.

Comments and Suggestions for Authors

Comment

This is an interesting study proving the applicability of a portable NIV device to mitigate exercise intolerance in COPD. The article is well written, despite the seemingly low number of participants the differences seem to be robust.

Response

We would like to thank the reviewer for his/her constructive comments and suggestions that have ultimately improved our manuscript. We have addressed all comments in the revised version of the manuscript. Revisions are shown as tracked changes.

Comments:

Comment

Patient characteristics. The authors should add drug therapy. Bronchodilators administered prior to exercise can reduce dynamic hyperinflation themselves. Were they encouraged/restricted to take their medication prior to exercise? Please, discuss.

Response

We appreciate the point made by the reviewer. Accordingly, on p.2 of the revised manuscript we indicate that: ‘patients were on optimal bronchodilator therapy including daily LAMA and LABA and no changes to medication were made during the trial.’

Comment

NIV. Why IPAP of 18 and EPAP of 8 cmH2O? Can these settings be modified? Pressures affect dynamic hyperinflation and I guess not all patients require the same settings. Please, discuss.

Response

We agree with the reviewer; this is a limitation we discussed with the manufacturer prior to the study; unfortunately it was not possible to allow modification of IPAP or EPAP. We have added the following comment to the discussion on p.12: “One technical limitation of the VItaBreath device is that the expiratory and inspiratory positive airway pressures (EPAP and IPAP, respectively) are fixed. Excessive EPAP can worsen hyperinflation and circulatory compromise. In our study population the majority of patients showed no worsening of DH or circulatory compromise whilst using pNIV during recovery periods. However in six patients (three per group) the improvement in DH was greater with PLB than pNIV, thereby suggesting that the fixed pressures were sub-optimal in at least some of our cohort. Accordingly, in future devices the ability to adjust EPAP and pressure support is desirable and could potentially be automated. “

Comment

The authors may consider removing Figure 2C as it does not add to the manuscript but may serve commercial benefit.

Response

We have removed Figure 2C as advised by this reviewer.

Comment

Table 1 and 2. Please provide p values for comparison.

Response

We have provided p values for both tables as requested.

Comment

Table 1. Please add units for the parameters

Response

We have added units in table 1 as requested